# Climate Data to Undertake Hygrothermal and Whole Building Simulations Under Projected Climate Change Influences for 11 Canadian Cities

**Abhishek Gaur \*, Michael Lacasse and Marianne Armstrong**

Construction Research Center, National Research Council Canada, 1200 Montreal Road,
Ottawa, ON K1A 0R6, Canada; Michael.Lacasse@nrc-cnrc.gc.ca (M.L.);
Marianne.Armstrong@nrc-cnrc.gc.ca (M.A.)
**\*** Correspondence: Abhishek.Gaur@nrc-cnrc.gc.ca; Tel.: +1-613-998-9799

**Abstract:** Buildings and homes in Canada will be exposed to unprecedented climatic conditions in the future as a consequence of global climate change. To improve the climate resiliency of existing and new buildings, it is important to evaluate their performance over current and projected future climates. Hygrothermal and whole building simulation models, which are important tools for assessing performance, require continuous climate records at high temporal frequencies of a wide range of climate variables for input into the kinds of models that relate to solar radiation, cloud-cover, wind, humidity, rainfall, temperature, and snow-cover. In this study, climate data that can be used to assess the performance of building envelopes under current and projected future climates, concurrent with 2 °C and 3.5 °C increases in global temperatures, are generated for 11 major Canadian cities. The datasets capture the internal variability of the climate as they are comprised of 15 realizations of the future climate generated by dynamically downscaling future projections from the CanESM2 global climate model and thereafter bias-corrected with reference to observations. An assessment of the bias-corrected projections suggests, as a consequence of global warming, future increases in the temperatures and precipitation, and decreases in the snow-cover and wind-speed for all cities.

**Dataset:** The full dataset can be accessed from: 10.17605/OSF.IO/UPFXJ.

**Dataset License:** CC0 1.0 Universal

**Keywords:** buildings; climate change; hygrothermal modelling; Canada; bias correction; climate model

---

## 1. Introduction

The impacts of climate change have been detected across Canada and other parts of the globe over the past 50–60 years [1,2]. For instance, long term temperature records across Canada suggest that since 1948 annual average temperatures in Canada have increased by 1.7 °C, approximately twice the average increase that was recorded globally in the same time-period [3,4]. Similarly, the average annual precipitation has increased in most parts of Canada since 1948 with increases of more than 30% recorded in the northernmost regions [4]. Other drastic changes in not only the means but also the extremes of climate elements have been recorded, such as decreases in the extent and duration of the permafrost, decreases in the extent of Arctic sea ice, decreases in the annual mean streamflow (particularly in southern Canada), early spring-melt driven riverine flows and floods, increased tendencies for drought, and increase (in some locations) in the intensities of cyclonic storms [3].

It is projected that by the end of the 21st century, annual average temperatures across Canada will increase by 1.8 °C for a low greenhouse gas emission scenario (RCP 2.6) to 6.3 °C for a high emission scenario (RCP 8.5) compared to the reference time-period of 1986–2005. The annual and winter precipitation is projected to increase in all parts of Canada; however, reductions in summer rainfall are projected for parts of southern Canada [4]. Climatic extremes are projected to become more intense and frequent as a consequence of climate change. For instance, it is projected that the annual highest daily temperature, which currently occurs every 20 years, will become a once in 5-years event by the mid-century under a low emission scenario and a once in 2-years event by the mid-century under a high emission scenario. Similarly, extreme precipitation amounts accumulated over a day or less are projected to increase in the future as a consequence of climate change [4]. As a result, buildings and infrastructures across Canada are expected to be exposed to drastically different annual and seasonal climatic conditions and more frequent and intense extreme events such as wildfires, flooding, wind-driven rain and heat spells over their design life. Failure to account for the non-stationarity of climate could lead to the premature failure of Canada's buildings and infrastructures.

Several studies in the past have adopted a building modelling based approach to evaluate the response of building envelopes to the anticipated effects of climate change. This is the case, for instance, [5] for the estimated potential moisture accumulation in the outmost layer of the façade of common wood frame wall constructions in Sweden under historical and future projected WDR loads, via performing hygrothermal simulations using a WUFI model. Historical (1961–1990) and future (2021–2050; 2071–2100) climate loads were obtained from 25 and 50 km resolution projections made by the a RCA3 regional climate model. [6] quantified the potential future impacts of climate change on antiquities stored in two historic buildings, one located in the Netherlands and the other located in Belgium. A high resolution (10 km) historical and future climate simulated by an RCM: RECO were used as inputs into an indoor air quality model (i.e., HAMBase) to simulate thermal and moisture conditions within the buildings. Increases in the indoor temperatures of up to 2 °C and increases in the indoor relative humidity of up to 2.6% in the first building and 1.8% in the second building were projected by the end of the 21st century, highlighting the potential moisture damage of artifacts kept in the buildings as a consequence of climate change. [7] analyzed the thermal performance of the Roland Levinsky Building in the UK under projected climate change influences. Future projections of climate obtained from the UKCP09 stochastic weather generator were used as inputs into an EnergyPlus V4.0 program to simulate the future thermal behavior of the building. The results obtained indicated that without additional cooling, the indoor temperatures are expected to rise in the building in the future. Several other building simulation-based studies have been performed to assess the impacts of climate change on the hygrothermal and thermal performance of building envelopes [8–13]. By undertaking whole building simulations, studies have also evaluated the energy behavior of buildings under current [14–17] and projected climate change influences [18–20].

The availability of reliable climatic data is central to the fidelity of the whole building simulations [21,22]. In Canada, the Canadian Weather Energy and Engineering Datasets (CWEEDS) and Canadian Weather Year for Energy Calculation (CWEC) weather files, produced by Environment and Climate Change Canada (ECCC; http://climat.meteo.gc.ca/prods_servs/engineering_e.html), are considered to be the standardized source of climate data required for performing building simulations for the historical time-period. The CWEEDS files provide hourly information on elements such as solar irradiance, temperature, dew point, pressure, wind-speed and direction, and other elements used for engineering purposes such as the building and solar system simulation and design. The CWEC files are subsets of the CWEEDS files and provide climate time-series for a typical meteorological year [23].

The data sources that provide access to weather files for climate change impact assessments of building envelopes include the University of Southampton's *CCWorldWeatherGen* tool (http://www.energy.soton.ac.uk/ccworldweathergen/), which provides access to historical and projected future weather files for 2100 locations distributed across the globe, of which 71 are located in Canada. The weather data is generated by combining climatic observations with the changes projected by the

HadCM3 climate model under the A2 greenhouse gas emission scenario discussed in the Special Report on Emission Scenarios [24].

　　　The objective of this research is to generate the reliable historical and projected future climatic data needed to undertake whole building and hygrothermal simulations under projected climate change influences for prominent Canadian cities distributed across Canada. The methodology used to prepare the datasets is described in Section 2, followed by an analysis and discussion on the key aspects of the data in Section 3; thereafter, conclusions are provided in Section 4.

## 2. Methods

### 2.1. Selected Cities

　　　A total of 11 cities located in different Canadian climate zones, and which are amongst the most populous urban centers in Canada, were selected for the climate data generation. The cities chosen for the analysis in this study are listed in Table 1 and shown in Figure 1.

**Table 1.** The Canadian cities for which climatic datasets were prepared.

| S. No. | City (SHORTNAME) | Latitude (°N) | Longitude (°E) | Population in 2016 (1000s) | Climate Zone |
|---|---|---|---|---|---|
| 1 | Calgary (CAL) | 51.0 | −114.1 | 1237.7 | Prairies/South British Columbia Mountains |
| 2 | Charlottetown (CHA) | 46.2 | −63.1 | 44.7 | Atlantic |
| 3 | Halifax (HAL) | 44.6 | −63.6 | 316.7 | Atlantic |
| 4 | Moncton (MNC) | 46.1 | −64.8 | 108.6 | Atlantic |
| 5 | Montreal (MON) | 45.5 | −73.6 | 1705.0 | Great lakes/St. Lawrence lowlands |
| 2 | Ottawa (OTT) | 45.4 | −75.7 | 934.2 | Great lakes/St. Lawrence lowlands |
| 3 | Saskatoon (SAS) | 52.1 | −106.7 | 246.4 | Prairies |
| 4 | St. Johns (STJ) | 47.6 | −52.7 | 206.0 | Atlantic |
| 5 | Toronto (TOR) | 43.7 | −79.4 | 2731.6 | Great lakes/St. Lawrence lowlands |
| 10 | Vancouver (VAN) | 49.3 | −123.1 | 2463.4 | Pacific coast/South British Columbia Mountains |
| 11 | Winnipeg (WIN) | 49.9 | −97.1 | 711.9 | Prairies |

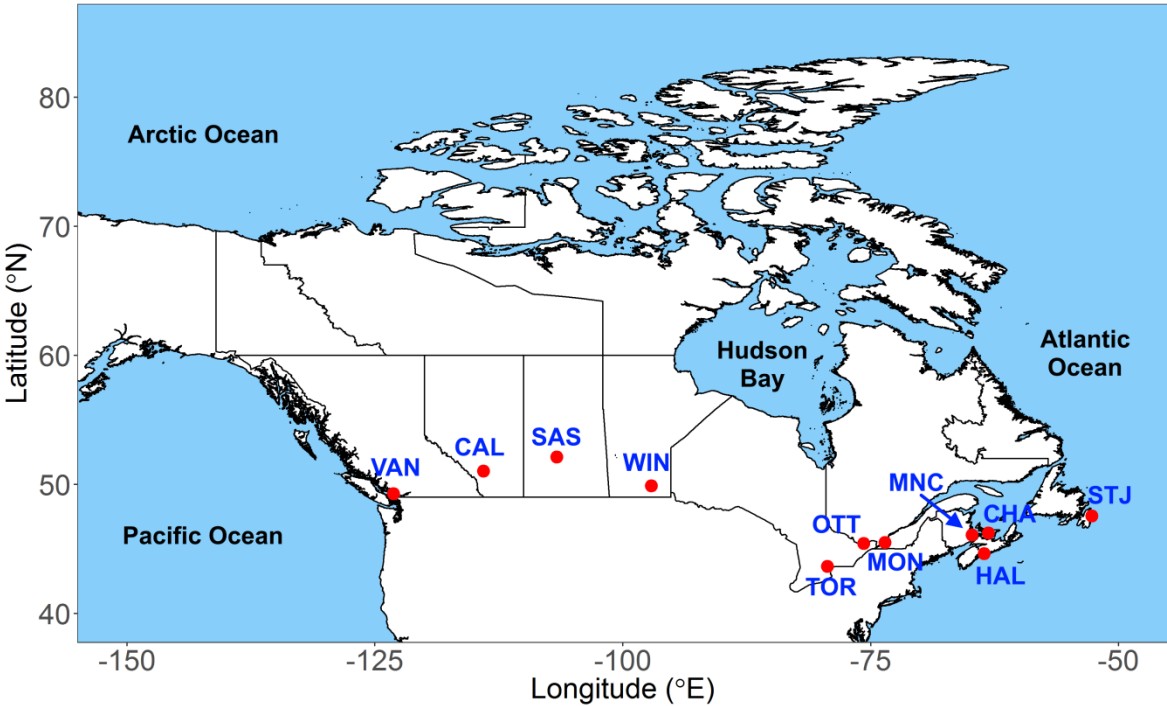

**Figure 1.** Location of cities selected for assessment in this study.

## 2.2. Generated Climate Variables

Hygrothermal and whole building simulations require high frequency climatic data comprising of a range of climatic variables. From a review of the climate data needs of common hygrothermal and whole building simulation software e.g., Hygric [25,26], Energy Plus [27], Delphin [28], and WUFI [29], and from extensive discussions with the experts working in this area, a set of climate variables, listed in Table 2, were identified as necessary and sufficient for undertaking both hygrothermal and whole building simulations. Furthermore, taking into consideration the availability of observational and climate model-based datasets, an hourly time-step was chosen for the climate data generation.

**Table 2.** Climate variables included as part of the datasets, their units, and source variables in the observational and CanESM2-LE databases.

| S. No. | Climate Variable | Units | Source (Observations) | Source (CanESM2-LE) |
|--------|------------------|-------|-----------------------|---------------------|
| 1 | Direct Horizontal Irradiance | $kJ/m^2$ | | |
| 2 | Diffused Horizontal Irradiance | $kJ/m^2$ | Hourly global horizontal irradiance | Hourly downward shortwave radiative flux |
| 3 | Direct Normal Irradiance | $kJ/m^2$ | | |
| 4 | Global Horizontal Irradiance | $kJ/m^2$ | | |
| 5 | Total Cloud Cover | % | Hourly total cloud cover | Hourly total cloud cover |
| 6 | Rainfall | mm | Hourly rainfall | Hourly precipitation, daily solid precipitation |

**Table 2.** *Cont.*

| S. No. | Climate Variable | Units | Source (Observations) | Source (CanESM2-LE) |
|--------|------------------|-------|-----------------------|---------------------|
| 7 | Wind direction | Degrees clockwise from north | Hourly wind direction | Hourly U and V components of wind |
| 8 | Wind speed | m/s | Hourly wind speed | |
| 9 | Relative humidity | % | Hourly relative humidity | Hourly relative humidity |
| 10 | Temperature | °C | Hourly temperature | Hourly temperature |
| 11 | Atmospheric Pressure | Pa | Hourly atmospheric Pressure | Hourly atmospheric Pressure |
| 12 | Snow-cover | 0 (no snow) or 1 (snow) | Daily snow depth | Daily snow depth |

*2.3. Methodology*

The methodology used to generate the climate data is summarized in Figure 2. The database of observational climate (described in Section 2.4.) was used to prepare a merged observational climate time-series spanning the baseline time-period (1986–2016) for each city. The database of climatic projections from CanRCM4 LE (described in Section 2.4.) was used to extract climatic projections for each city spanning the baseline and future time-periods (specified in Section 2.4.). Merged observations and CanRCM4 LE projections for the baseline time-period were used to derive bias-correction factors, which were used to bias-correct the climate projections and obtain bias-corrected climate time-series over the baseline and future time-periods.

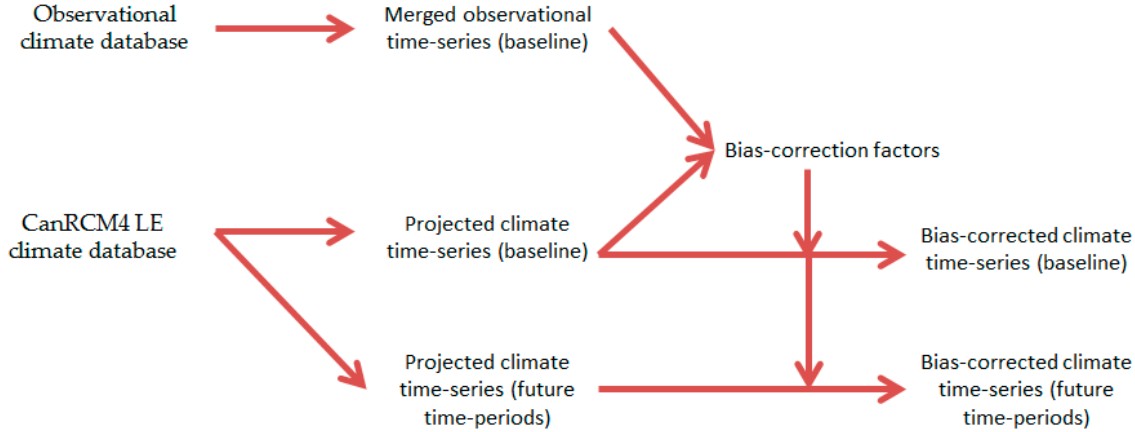

**Figure 2.** A summary of the methodology used to generate climate data for each city.

*2.4. Data Sources*

2.4.1. Climate Observations

Hourly observations of the near surface global horizontal irradiance, total cloud cover, rainfall, relative humidity, air temperature, station pressure, 10 m wind-speed and wind-direction, and daily observations of the snow-depth were collected for all of the Environment and Climate Change Canada (ECCC) climate gauging stations located within the domains of the selected cities and containing more than 1 year of data between the time-period: 1986–2016 (hereafter referred to as the baseline time-period). The daily observations of the snow-depth were considered instead of hourly observations because of their greater availability and reliability as compared to hourly observations (Robert Morris, ECCC, personal communication). The climate gauging stations where the observational data were collected are summarized in Tables S1–S11 for the 11 cities listed in Table 1. The station identifiers

from ECCC, such as the program source, element, station name and climate ID of the stations, their geographical locations, and the time-period of the data collection over the baseline period, have been provided.

2.4.2. Climate Model Simulations

The historical and projected future climate data are based on the CanESM2 large ensemble (CanESM2 LE) simulations [30]. CanESM2 is a Global Climate Model (GCM) having an interactive atmosphere, ocean, sea ice, land, and carbon cycle components completed at a geographical horizontal spacing of ~2.8° [31]. A large initial condition ensemble of CanESM2, consisting of 50 simulations, was randomly initialized starting on 1 January 1950 from the 5 historical ensemble members, by selecting 10 random sets of cloud physics parameterizations in the model [32]. The historical simulations covered the time-period 1950–2004, and the RCP8.5 scenario [33] was used to extend the simulations to the end of the 21st century. To dynamically downscale the CanESM2 LE to a 0.44° grid, regional simulations were performed with the CanRCM4 RCM [34]. A subset of the CanRCM4 dynamically downscaled CanESM2 LE simulations (referred as CanRCM4 LE hereafter), comprising of 15 realizations, have been archived in hourly time-steps by ECCC and were acquired for the purposes of this study.

The climate data consisted of the variables listed in Table 2 and encompassed the time-period 1950–2100. Three 31-year long time-series corresponding to a baseline and two future time-periods coincident with globally averaged future warmings of 2 °C and 3.5 °C were extracted from the database. A duration of 31 years was chosen so that the analyzed impacts could be attributed to the climate as opposed to being attributed to short-term weather changes. This is in line with the recommendations made by the World Meteorological Organization (WMO) on the use of at least 30 years of data for climate change impact assessments [35]. An analysis of the CanRCM4 LE simulations in [30] indicated that the above prescribed levels of global warming will be reached in the future over the 31-year time-periods, respectively of 2034–2064 and 2062–2092. As such, 15 members of the hourly CanESM2 LE projections covering the baseline and the aforementioned future time-periods were extracted for the grids covering the selected cities for the generation of historical and projected time-series of the climate variables. A single RCM grid was found to sufficiently encompass almost all of the cities considered for this analysis, and therefore a combination of projections from more than one RCM grid was not performed in this study.

*2.5. Derived Variables*

The following variables were derived as they were unavailable directly in either the observational or climate model projections:

**Rainfall**

Rainfall (R) values were available directly in the observational climatic database but unavailable in the CanRCM4 LE database. Therefore, the CanRCM4 LE projections of rainfall were derived from the values of the total precipitation (P) and total solid precipitation (SP), using Equation (1):

$$R = P - SP \tag{1}$$

**Snow-cover**

In respect to both the observations and CanRCM4 LE projections, the snow-cover flags constituting the values of 0 (no-snow) and 1 (snow) were derived from the values of the snow-depth, using Equation (2):

$$S_c = \left\{ \begin{array}{l} 0 \text{ for } S_d \leq 0 \\ 1 \text{ for } S_d > 0 \end{array} \right\} \tag{2}$$

**Direct horizontal, direct normal, and diffused horizontal solar radiation**

With reference to both the observations and climate model projections, the values of the direct horizontal irradiance (DRI), direct normal irradiance (DNI), and diffused horizontal irradiance (DHI)

were derived from the hourly values of the surface global horizontal irradiance (GHI). The methodology used is based on [14] and involves the following steps:

- The horizontal components of the hourly extra-terrestrial solar radiation (Xtr) were calculated, and their ratio (Xtr/GHI) was calculated to obtain the clearness index ($k_T$).
- The values of $k_T$ were used to calculate the diffused fraction $k_d$ using the method of [36], described in Equation (3):

$$k_d = \begin{cases} 1 - 0.249k_T & \text{for } k_T < 0.35 \\ 1.557 - 1.84k_T & \text{for } 0.35 \leq k_T < 0.75 \\ 0.177 & \text{for } k_T \geq 0.75 \end{cases} \tag{3}$$

- The values of DHI and DRI were calculated using Equations (4) and (5) respectively:

$$\text{DHI} = \text{GHI} \times k_d \tag{4}$$

$$\text{DRI} = \text{GHI} - \text{DHI} \tag{5}$$

- The values of DNI were calculated using Equation (6):

$$\text{DNI} = \frac{\text{DRI}}{\sin(\theta)} \tag{6}$$

where $\theta$ denotes the solar elevation angle.

### 2.6. Bias-correction of Climate Model Data

Many studies have found that climate projections made by Regional Climate Models (RCMs) are associated with bias. For instance, [37] compared the precipitation and temperature values simulated by an RCM: Reg-CM3 with spatially averaged observations to quantify the bias associated with the RCM. In this study, CanRCM4 LE simulations were bias-corrected with reference to a spatially averaged observational climatic data which was prepared by combining station level climate observations recorded at all climate gauging stations located within the city domains (listed in Tables S1–S11 of the supplementary material).

Many bias-correction methods have been used to correct climate data from global and regional climate models. The methods used range from simple scaling and additive corrections [38,39] to more advanced histogram equalization [40–42] and multivariate [43] bias correction methods.

The following process was used to bias correct the CanRCM4 LE simulation results in this study:

(i)　The temperature, relative humidity, wind speed, wind direction, pressure, and cloud-cover were corrected using bias correction factors that capture the differences between the modelled and observed means of these variables for each hour and month combination. By comparing the merged observational and CanRCM4 LE climate data, the additive correction factors were derived for the temperature, relative humidity, wind direction, and cloud-cover, whereas the multiplicative correction factors were derived for the wind speed and pressure. The derived bias-correction factors were later added (in the case of the temperature, relative humidity, wind direction, and cloud-cover) and multiplied (in the case of the wind speed and pressure variables) with CanRCM4 LE projections to obtain the bias corrected projections.

(ii)　With respect to the rainfall, the bias associated with the total number of wet hours and magnitudes of the rainfall events simulated by CanRCM4 LE were corrected. The bias correction factors were derived for each month, and only days with no missing observational rainfall values were considered for the estimation of the bias-correction parameters. It was noted that CanRCM4 LE tended to have a negative wet hour bias i.e., the number of wet hours simulated by the model were greater than the observations; this is similar to the bias evident for many other RCMs [37,38]. The procedure adopted to correct this bias was as follows [38]:

- The observational and model rainfall intensities were sorted in a decreasing order.
- The point at which the observations reached above zero provided the corresponding threshold intensity $R_{th}$ of the CanRCM4 LE model.
- The $R_{th}$ is then subtracted from each of the hourly rainfall values and truncated at zero so that the numbers of dry hours are identical for both the model and observational data.
- The corrected hourly rainfall is then:

$$R_{dd}(t) = \max(R_{\text{mod}}(t) - R_{th}, 0) \tag{7}$$

Following the wet hour number bias correction, the bias of the rainfall magnitude simulated by the model is corrected using the ratio of the rainfall present in the observations and using the ratio simulated by the model:

$$R_{bcs}(t) = R_{dd}(t) \times \frac{\overline{R_{obs}}}{\overline{R_{dd}}} \tag{8}$$

where $\overline{R_{obs}}$ and $\overline{R_{dd}}$ denote, respectively, the average rainfall in the observations and the wet hour corrected modelled rainfall.

(iii)　To obtain the bias-corrected CanRCM4 LE snow-cover data, corrections were made to the CanRCM4 LE derived snow-depths that were later converted to the snow-cover using Equation (2). The bias correction factors were derived for each month, and only days with no missing observational snow-depth values were considered for the estimation of the bias-correction parameters. When a negative bias was associated with the total number of simulated snow hours, the procedure used to correct the negative bias in the total numbers of wet rainfall hours was used for the bias-correction of the snow hours. When a positive bias was associated with the total number of simulated snow hours, i.e., the numbers of snow hours simulated by the model were smaller than the observations, the number of wet hours in CanRCM4 LE needed to be increased. The increase in, for example, the $N_{wd}$ number of wet hours was introduced by first calculating the 24-h running mean of the snow-depth using the CanRCM4 snow-depth data, and thereafter converting the $N_{wd}$ no-snow events with the highest 24-h snow depths into the snow events.

(iv)　With regard to the solar radiation, multiplicative bias-correction factors were initially derived for each month and hour combination (as in 2.6i); however, it was noted that the bias-correction factors for the early morning and late evening hours were not stable due to the lack of non-zero solar data for those hours. Another observation made from the initial sets of bias-correction factors derived for each hour and month was that their values were consistently greater than 1 (i.e., the model under-predicted the global solar radiation) for the hours until noon and under 1 (i.e., the model over-predicted the global solar radiation) for the hours after noon. Therefore, in an effort to obtain more robust sets of bias-correction factors, two bias-correction factors were derived for each month: one was derived for the hours 0–12 and another was derived for the hours 13–23.

## 3. Analysis and Discussion

### 3.1. Validation of Bias Correction Procedure

The reliability of the bias-correction procedure used to correct the climatic variables in this study was tested using a split sample approach. For each city and climate variable, the dates on which complete sets (with no missing values) of climatic data were available were recorded. The bias-correction parameters were derived using the climatic data recorded for the first half of the dates (calibration time-period) and used to predict the bias-corrected values over the second half of the dates (validation time-period). Because the time-period of data availability varied for different climate

variables in different cities, the length and duration of the calibration and validation time-periods differed for different climate variables and for different cities. The predicted climate variables over the validation time-period were compared with the climatic observations to assess the accuracy of the predictions and the reliability of the procedure used to perform the bias correction. As the intention was to test the reliability of the bias-correction procedure, the validation exercise was only performed on one (r1) of the fifteen hourly realizations of climate projections available from CanRCM4 LE.

The validation results for the pressure, relative humidity, temperature, cloud-cover, wind speed, and wind direction are summarized in Table S12 of the supplementary material. The values of the differences in the percent root mean squared error (RMSE) associated with the raw model results and bias-corrected model results ($RMSE_{model}$ (%) − $RMSE_{bc}$ (%)) are provided. The values are found to be greater than 0 for almost all of the cases, indicating that the accuracy of the CanRCM4 LE projections improved in terms of RMSE, as a result of the bias-correction procedure. Furthermore, the magnitude of the mean bias associated with the CanRCM4 LE projections ($MB_{model}$ (%)) and bias-corrected projections ($MB_{bc}$ (%)) are also provided. It can again be noted that the bias-associated CanRCM4 LE projections have a lower (in absolute terms) bias associated with them when compared to the raw CanRCM4 LE results. Table S12 of the supplementary material also provides the average values of these variables over the validation time-period, calculated from the observational data ($Obs_{avg}$), CanRCM4 LE projections ($Model_{avg}$), and bias corrected CanRCM4 LE projections ($Model_{bc,avg}$). Again, it is noted that the bias-corrected values of CanRCM4 LE run1 are closer to the observational values when compared to the uncorrected values.

A summary of the validation results for the rainfall and snow-cover variables is presented in Table S13 of the supplementary material. The values of the mean wet-day rainfall, total number of rainy days, and total number of snowy days obtained from the model simulations, bias-corrected model simulations, and observational records are presented. It is noted that the mean wet day rainfall is consistently under-predicted in the CanRCM4 LE results, whereas the total number of rainy hours are over-predicted. The results from the validation procedure indicate that these biases are addressed by the bias correction procedure adopted in this study. The sign of bias in the total number of simulated hours with snow-cover is found to be not as consistent as in the number of rainy hours; however, in most cities (except Charlottetown) the snow hours are over predicted. The bias correction procedure used in this study is able to eliminate this bias with a reasonable accuracy, as seen in the results presented in Table S13 of the supplementary material.

The validation results for the global horizontal irradiance (GHI) are presented in Table S14 of the supplementary material and Figure 3. In Table S14, the difference in RMSE, associated with the uncorrected and corrected CanRCM4 LE run1 results ($RMSE_{model}$ (%) − $RMSE_{bc}$ (%)), and the percent mean bias, associated with the uncorrected ($MB_{model}$ (%)) and corrected ($MB_{bc}$ (%)) CanRCM4 LE run1 results, are presented. An improvement in the accuracy of the GHI projections in terms of RMSE is obtained for all cities. The mean bias values suggest an over-prediction of the global solar radiation values by the model, which is reduced as a result of the bias-correction procedure. The bias correction of GHI also corrects the hourly distribution of the derived solar variables, for instance the Direct Normal Irradiance (DNI). It is expected that over the course of the day, the DNI values increase gradually after sunrise, peak around noon, remain high in the afternoon, and then recede gradually until sunset when they become zero. The hourly distribution of the average DNI over the validation period before and after the bias-correction is shown in Figure 3 for all cities. Due to a combination of positive bias for the model simulated GHI values and low solar elevation angles in the evening, the DNI values obtained from the uncorrected CanRCM4 LE data peaked in the evening instead of around noon, as seen in Figure 3 (left). After the bias correction of the GHI values, the derived hourly DNI values are correctly distributed across the day.

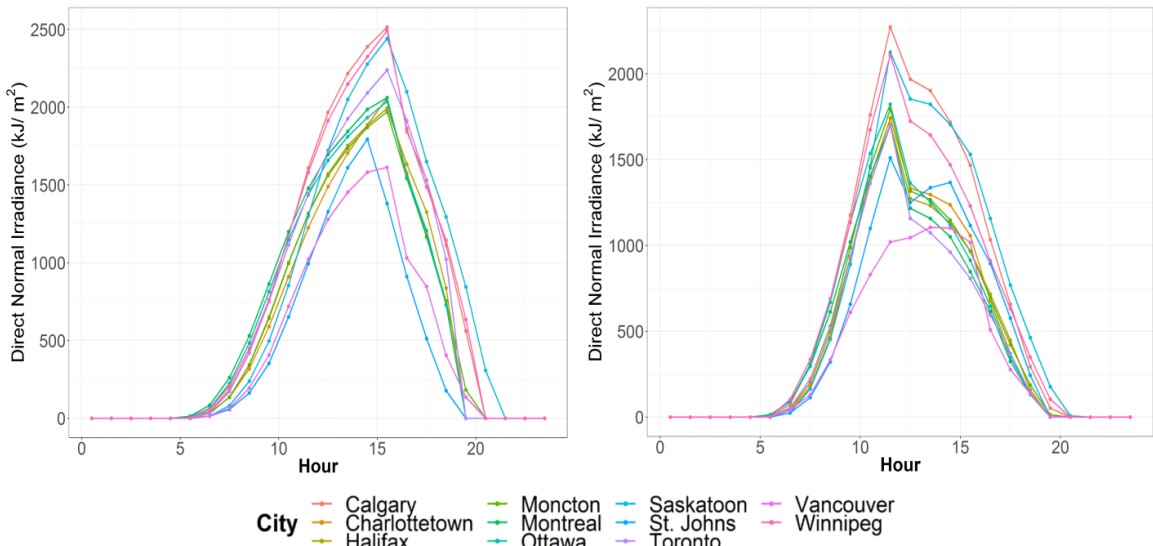

**Figure 3.** The hourly distribution of the mean Direct Normal Irradiance (DNI) over the validation time-period, obtained from the Global Horizontal Irradiance from CanRCM4 LE run1 before (left) and after (right) the bias correction.

*3.2. Projected Changes in Climate Under 2 °C and 3.5 °C Global Warming*

The projected changes obtained from the bias-corrected CanRCM4 LE projections under 2.0 °C and 3.5 °C of global warming are summarized in Tables 3 and 4 respectively. Statistics are provided of the percent changes in the mean (over the 31 year time-period) global horizontal irradiance (GHI), total cloud cover (TCC), total rainfall (RAIN), wind direction (WDIR), wind speed (WSP), relative humidity (RHUM), temperature (TEMP), atmospheric pressure (ATMPR), and total hours with snow-cover (SNOW HOURS) projected for all cities. In addition to the realization averaged changes (averaged across the fifteen model realizations), the range of the projected changes have also been provided. It is noted that temperature is projected to increase across all the cities under both 2 °C and 3.5 °C of global warming. The degree of increase projected for the cities, in all cases, is higher than the global average values of 2 °C and 3.5 °C in all cases. In some instances, especially for cities located in the prairies such as Saskatoon and Winnipeg, the degree of projected increases in temperature is determined to be as high as about twice the globally averaged projected increases in temperatures. The projected changes in temperatures across all 11 cities is also evident from Figure 4, where average temperatures over the baseline time-period, and under global warming of 2 °C and 3.5 °C, are shown. The relative humidity is projected to increase, whereas wind speeds are projected to decrease in all cities. The relatively smaller and uncertain changes in GHI, TCC, WDIR, and ATMPR are also noted. Rainfall is projected to increase in all cities, which is also evident from Figure 5, where annual rainfall, averaged across all 15 bias-corrected CanRCM4 LE runs, are shown for the baseline time-period and under projected global warming of 2 °C and 3.5 °C. Finally, drastic reductions in the snow-cover hours are obtained for all cities as a consequence of global warming.

In addition to the projected mean changes, uncertainty in the projected changes as a consequence of the internal climate variability is also evident from the results presented in Tables 3 and 4. It can be seen that the range of projected changes from the 15 runs, as evidenced from the maximum and minimum values of the projected changes, are very high and hence should clearly not be ignored when performing climate change impact assessments.



**Table 3.** Projected changes in climatic variables under a global warming of 2 °C with reference to climate modelled over baseline time-period.

| Statistic | GHI (%) | TCC (%) | RAIN (%) | WDIR (°) | WSP (%) | RHUM (%) | TEMP (°C) | ATMPR (%) | SNOW HOURS (%) |
|---|---|---|---|---|---|---|---|---|---|
| Calgary | | | | | | | | | |
| Mean | −0.9 | 0.6 | 16.4 | −0.9 | −3.1 | 1.0 | 2.7 | 0.1 | −21.7 |
| Max | 1.2 | 2.0 | 24.9 | 0.6 | −0.1 | 1.9 | 3.2 | 0.1 | 11.9 |
| Min | −2.7 | −0.7 | 1.1 | −2.1 | −6.0 | −0.3 | 2.1 | 0.1 | −44.3 |
| Charlottetown | | | | | | | | | |
| Mean | 1.5 | −1.3 | 8.7 | −0.9 | −1.2 | 0.2 | 2.9 | 0 | −48.8 |
| Max | 2.9 | -0.4 | 14.7 | 0.3 | −0.4 | 0.6 | 3.3 | 0.1 | −39.8 |
| Min | 0.4 | −2.2 | 2.5 | −3.7 | −2.5 | 0 | 2.5 | 0 | −64.8 |
| Halifax | | | | | | | | | |
| Mean | 0.5 | 0.2 | 6.3 | −0.9 | −1.2 | 0.9 | 2.8 | 0 | −65.6 |
| Max | 1.6 | 1.0 | 10.2 | 0.5 | −0.5 | 1.2 | 3.2 | 0.1 | −56.0 |
| Min | −0.4 | −0.4 | 2.4 | −3.7 | −2.5 | 0.6 | 2.4 | 0 | −79.6 |
| Moncton | | | | | | | | | |
| Mean | 0.1 | 0.2 | 10.5 | −0.7 | −0.8 | 1.6 | 2.9 | 0 | −45.4 |
| Max | 2.0 | 1.0 | 17.1 | 0.9 | 0 | 2.1 | 3.4 | 0.1 | −37.1 |
| Min | −0.8 | −1.1 | 3.0 | −2.4 | −2.3 | 1.2 | 2.6 | 0 | −60.3 |
| Montreal | | | | | | | | | |
| Mean | 0.5 | 0.4 | 9.4 | −0.7 | −1.0 | 0.7 | 3.1 | 0 | −40.8 |
| Max | 2.2 | 1.4 | 14.8 | 0.9 | 0.6 | 1.3 | 3.6 | 0 | −30.0 |
| Min | −0.5 | −0.8 | 1.7 | −3.3 | −1.9 | 0.1 | 2.6 | 0 | −52.7 |
| Ottawa | | | | | | | | | |
| Mean | 0.3 | 0.5 | 8.1 | −0.4 | −1.1 | 0.9 | 3.1 | 0 | −33.5 |
| Max | 1.9 | 1.5 | 15.4 | 1.6 | 0.2 | 1.6 | 3.6 | 0 | −24.4 |
| Min | −1.1 | −0.5 | 1.9 | −2.9 | −2.0 | 0.3 | 2.5 | 0 | −44.9 |
| Saskatoon | | | | | | | | | |
| Mean | −1.0 | 0.9 | 10.2 | −2.1 | −2.3 | 0.5 | 3.1 | 0.1 | −41.9 |
| Max | 0 | 1.7 | 18.3 | 0 | −0.1 | 1.4 | 3.8 | 0.1 | −28.2 |
| Min | −2.1 | −0.2 | −1.6 | −3.5 | −4.0 | −0.5 | 2.5 | 0 | −53.9 |
| St. Johns | | | | | | | | | |
| Mean | 1.1 | −0.3 | 4.0 | 0.2 | −1.1 | 0.2 | 2.6 | 0 | −62.4 |
| Max | 3.5 | 0.8 | 11.4 | 2.0 | −0.1 | 0.9 | 3.0 | 0.1 | −50.7 |
| Min | −1.1 | −1.0 | −4.4 | −1.4 | −2.2 | −0.1 | 2.3 | 0 | −74.1 |
| Toronto | | | | | | | | | |
| Mean | 1.2 | 0.2 | 3.1 | −1.0 | −2.8 | 0.1 | 2.7 | 0 | −47.7 |
| Max | 2.3 | 1.0 | 14.5 | 1.2 | −1.4 | 0.8 | 3.1 | 0 | −39.2 |
| Min | 0.2 | −1.1 | −7.1 | −3.5 | −3.5 | −0.5 | 2.3 | 0 | −58.4 |
| Vancouver | | | | | | | | | |
| Mean | 1.3 | 0 | 3.9 | −1.0 | −3.0 | 0.5 | 2.6 | 0 | −99.6 |
| Max | 2.9 | 1.5 | 12.0 | 1.3 | −0.5 | 1.0 | 2.9 | 0 | −97.5 |
| Min | −0.5 | −1.6 | −2.4 | −4.3 | −4.9 | 0 | 2.2 | 0 | −100.0 |
| Winnipeg | | | | | | | | | |
| Mean | −1.2 | 0.7 | 6.7 | −2.0 | −1.3 | 0.2 | 3.3 | 0 | −20.3 |
| Max | −0.4 | 1.8 | 17.2 | 0.7 | 0.2 | 1.2 | 4.0 | 0.1 | −32.1 |
| Min | −2.0 | −0.5 | −7.2 | −3.4 | −3.2 | −0.7 | 2.8 | 0 | −51.9 |

**Table 4.** Projected changes in the climatic variables under a global warming of 3.5 °C with reference to the climate modelled over the baseline time-period.

| Statistic | GHI (%) | TCC (%) | RAIN (%) | WDIR (°) | WSP (%) | RHUM (%) | TEMP (°C) | ATMPR (%) | SNOW HOURS (%) |
|---|---|---|---|---|---|---|---|---|---|
| | | | | Calgary | | | | | |
| Mean | −1.3 | 1.0 | 22.7 | −1.8 | −5.4 | 1.4 | 4.8 | 0.2 | −45.3 |
| Max | 0.1 | 2.6 | 34.3 | 0.7 | −3.6 | 3.0 | 5.2 | 0.2 | −33.3 |
| Min | −3.8 | 0 | 9.8 | −3.8 | −7.8 | 0.4 | 4.5 | 0.2 | −56.3 |
| | | | | Charlottetown | | | | | |
| Mean | 1.6 | −1.4 | 13.4 | −1.9 | −2.6 | 0.5 | 4.8 | 0 | −74.4 |
| Max | 2.8 | −0.5 | 19.6 | −0.2 | −1.7 | 0.9 | 5.2 | 0.1 | −63.0 |
| Min | 1.0 | −2.0 | 7.2 | −4.1 | −4.3 | 0.1 | 4.3 | 0 | −86.6 |
| | | | | Halifax | | | | | |
| Mean | 0 | 0.7 | 9.7 | −2.3 | −2.6 | 1.7 | 4.7 | 0.1 | −87.2 |
| Max | 0.8 | 1.7 | 16.7 | 0 | −1.5 | 1.9 | 5.1 | 0.1 | −79.8 |
| Min | −1.1 | 0 | 2.6 | −4.1 | −3.8 | 1.4 | 4.4 | 0 | −92.7 |
| | | | | Moncton | | | | | |
| Mean | −0.5 | 0.7 | 17.5 | −1.7 | −2.0 | 3.1 | 5.0 | 0 | −76.5 |
| Max | 0.3 | 1.6 | 23.0 | 0.4 | −0.7 | 3.5 | 5.4 | 0.1 | −66.0 |
| Min | −1.6 | 0 | 10.1 | −3.1 | −3.1 | 2.7 | 4.6 | 0 | −84.9 |
| | | | | Montreal | | | | | |
| Mean | −0.1 | 1.0 | 16.8 | −2.2 | −2.4 | 1.5 | 5.2 | 0 | −68.9 |
| Max | 0.6 | 1.8 | 25.3 | −0.9 | −1.3 | 1.9 | 5.6 | 0 | −60.6 |
| Min | −0.8 | 0.1 | 9.6 | −3.7 | −3.6 | 1.0 | 4.8 | 0 | −76.1 |
| | | | | Ottawa | | | | | |
| Mean | −0.5 | 1.2 | 14.4 | −1.8 | −2.2 | 1.8 | 5.2 | 0 | −62.5 |
| Max | 0.5 | 1.8 | 21.0 | 0.7 | −1.0 | 2.3 | 5.6 | 0 | −53.4 |
| Min | −1.5 | 0.5 | 5.7 | −4.3 | −3.5 | 1.2 | 4.8 | 0 | −73.8 |
| | | | | Saskatoon | | | | | |
| Mean | −1.8 | 1.6 | 15.5 | −3.7 | −4.2 | 0.8 | 5.4 | 0.1 | −22.6 |
| Max | −0.3 | 3.4 | 32.3 | −2.1 | −2.5 | 2.3 | 5.8 | 0.1 | −8.0 |
| Min | −3.2 | 0.8 | 6.9 | −6.1 | −6.0 | −0.3 | 5.0 | 0 | −41.8 |
| | | | | St. Johns | | | | | |
| Mean | 2.7 | −0.7 | 7.6 | −0.1 | −2.1 | 0.3 | 4.7 | 0 | −89.3 |
| Max | 4.4 | 0.3 | 15.7 | 2.6 | −1.4 | 1.0 | 5.0 | 0.1 | −77.1 |
| Min | 0.9 | −1.5 | 1.8 | −2.2 | −2.8 | 0 | 4.3 | 0 | −96.2 |
| | | | | Toronto | | | | | |
| Mean | 0.8 | 1.0 | 6.5 | −2.6 | −5.1 | 0.5 | 4.7 | 0 | −76.2 |
| Max | 1.8 | 1.6 | 15.3 | −0.2 | −3.6 | 1.0 | 4.9 | 0 | −69.7 |
| Min | −0.1 | −0.1 | 0.5 | −4.4 | −6.3 | −0.1 | 4.3 | 0 | −82.7 |
| | | | | Vancouver | | | | | |
| Mean | 2.5 | −0.4 | 4.4 | −2.0 | −5.8 | 0.5 | 4.3 | 0 | −100.0 |
| Max | 3.9 | 0.6 | 9.7 | −0.6 | −4.1 | 1.1 | 4.8 | 0 | −100.0 |
| Min | 0.7 | −1.8 | −2.8 | −4.5 | −7.7 | 0.2 | 4.0 | 0 | −100.0 |
| | | | | Winnipeg | | | | | |
| Mean | −2.7 | 1.8 | 14.6 | −2.8 | −2.7 | 0.7 | 5.6 | 0 | −41.5 |
| Max | −1.5 | 2.7 | 29.0 | −0.6 | −0.5 | 2.0 | 6.0 | 0.1 | −32.1 |
| Min | −3.4 | 0.6 | −1.8 | −4.7 | −4.4 | 0 | 5.3 | 0 | −51.9 |

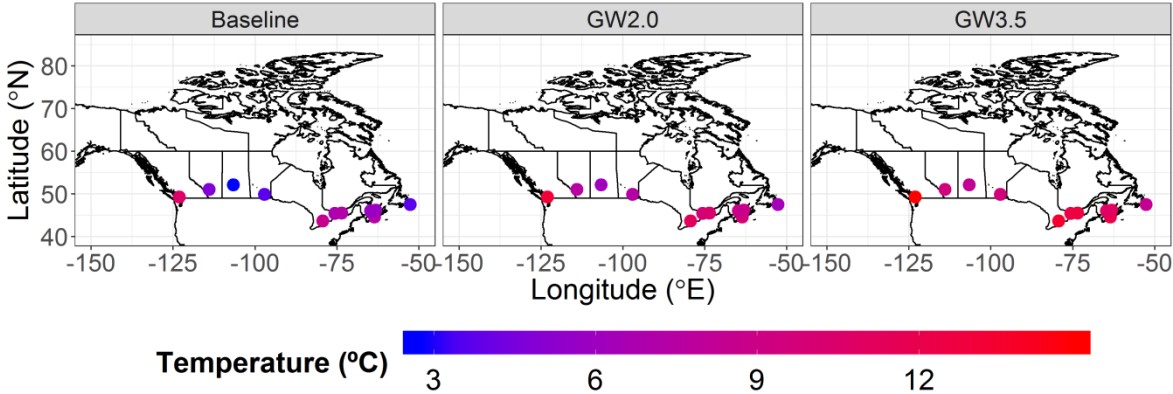

**Figure 4.** Mean temperatures in the selected cities over the baseline time-period, and under projected global warmings of 2.0 °C and 3.5 °C.

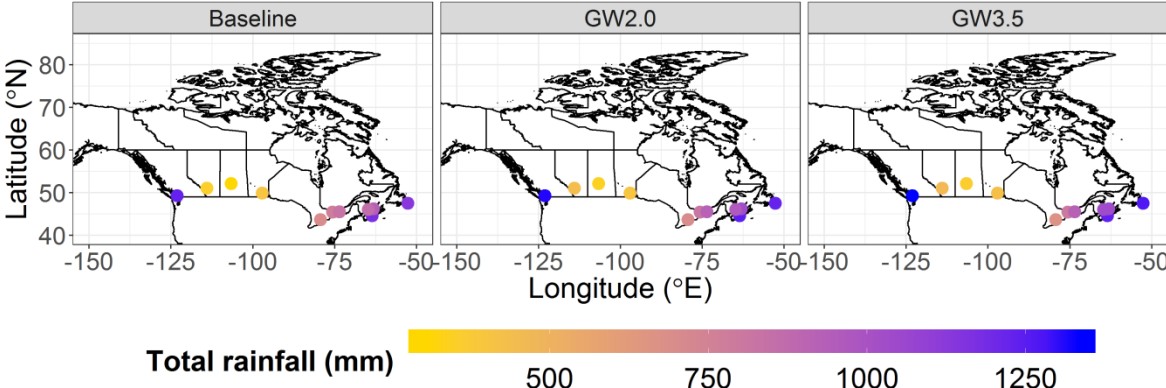

**Figure 5.** Mean annual total rainfall in the selected cities over the baseline time-period, and under projected global warmings of 2.0 °C and 3.5 °C.

## 4. Conclusions

Many studies have discussed the drastic consequences that climate change will have on the Canadian climate. Buildings and civil infrastructures residing in different Canadian cities needs to be designed and evaluated under projected climates to which they are likely to be exposed to in the future. Reliable future projections of the climate are required for undertaking hygrothermal and whole building simulations, and these projections should permit the evaluation of the performance of building envelopes under current and future climates.

This study generates the climatic data necessary to undertake building simulations for 11 cities located across Canada. The data is comprised of hourly time-series of direct horizontal irradiance, diffused horizontal irradiance, direct normal irradiance, global horizontal irradiance, total cloud cover, rainfall, wind direction, wind speed, relative humidity, temperature, atmospheric pressure, and snow-cover for a baseline time-period spanning 1986–2016 and 31-year long future time-periods when global warming of 2 °C and 3.5 °C (with reference to the baseline time-period) is expected to be reached in the future. The climatic datasets that were generated as part of this study capture the effects of the internal variability of the climate on future climate projections as the fifteen hourly realizations that are part of the datasets are derived from the large ensemble of climates simulated by the Canadian Regional Climate Model—version 4 (CanRCM4), each initialized under a different set of initial conditions in the CanESM2 global climate model. The CanRCM4 LE projections are evaluated for the presence of bias by comparing them with observations over the historical time-period. The bias-correction procedure that was used was evaluated and the results indicate that the bias associated with the CanRCM4 LE projections are reduced by the bias correction step. An analysis of the bias-corrected time-series of the various climate elements indicates that the temperature and rainfall are expected to increase and that

the snow-cover is expected to decrease as a consequence of future increases in global temperatures. This is in line with the findings made by many previous studies that have evaluated the future impacts of climate change on the Canadian climate [3,4,44]. The above discussion suggests that the generated datasets provide building practitioners with a reliable source of climate data for evaluating the thermal and hygrothermal response of building envelopes under projected climate change influences in several Canadian cities.

The generated datasets do not, however, capture the uncertainty in climatic projections that may arise from the use of other state-of-the-art GCMs and RCMs in simulating the regional climate and future climate projections. The bias-correction procedure used to generate the datasets corrects for monthly and/or hourly means of the climate variables instead of the entire distributions, which may result in an underestimation of the extremes of these variables in the bias-corrected data [40,41]. Furthermore, the cross-correlation structure between the climate variables may not be well preserved in the bias-corrected data [43]. Improving these datasets, via some of the above themes, is the future direction of this work.

**Supplementary Materials:** The following are available online at http://www.mdpi.com/2306-5729/4/2/72/s1, File: data-498712_supplementary material.docx.

**Author Contributions:** Conceptualization, M.L. and M.A.; methodology, A.G.; software, A.G.; validation, A.G.; formal analysis, A.G.; investigation, A.G.; resources, M.L. and M.A.; data curation, A.G.; writing—original draft preparation, A.G.; writing—review and editing, M.L.; visualization, A.G.; supervision, M.L.; project administration, M.A.; funding acquisition, M.A.

**Funding:** Funding for this research came from Infrastructure Canada as part of the Climate-Resilient Buildings and Core Public Infrastructure (CRB-CPI) project.

**Acknowledgments:** The authors would like to thank researchers from Façade Systems and Products group of National Research Council Canada and especially Abdelaziz Laouadi for providing feedback on the layout of the weather files. Observational climate data and CanRCM4 projections were obtained courtesy of Environment and Climate Change Canada. Discussions about these datasets with Robert Morris and Alex Cannon greatly improved our understanding of them. Comments from two anonymous reviewers helped improve the quality and readership of this paper.

**Conflicts of Interest:** The authors declare no conflict of interest.

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
