# Peer review of "Climate Data to Undertake Hygrothermal and Whole Building Simulations Under Projected Climate Change Influences for 11 Canadian Cities"

_data, 2019_

Reviewer 1 Report

This Reviewer would like to thank the authors for sharing their valuable work with the scientific community.

The manuscript is well structured and the study was carried out with adequate scientific methodologies. In the opinion of this Reviewer  the manuscript deserves to be published on Data.

This Reviewer  raises only the following couple of issues/requests of clarification.

First, it could be better to insert at the beginning of Section 3 an outline about the methodology diagrams flows (how many steps, the aim of each step, the actors involved in each step, etc.); maybe the use of UML or SysML could help authors describing the proposed system view in a more structured fashion.

The current trend of several recent scientific studies (e.g., http://dx.doi.org/10.1109/CDC.2014.7040273, https://doi.org/10.3390/su7078782, https://doi.org/10.1109/SMC.2018.00242, https://doi.org/10.3390/en10010117, documents that could be cited in the text) related to energy behaviour of buildings focuses on the deployment of renewable energy sources (for instance to achieve zero energy building) and of storage systems. Actually, the fidelity level of whole building simulation relies on the accuracy of distributed generation forecast, which in turn depends on the availability of weather forecast (https://doi.org/10.1016/j.renene.2011.05.033,  https://doi.org/10.3390/en12010100, documents that could be cited in the text). It consequently seems to this Reviewer that climate data that can be used to assess also the performance of building energy systems, under current and projected future climates. The authors could analyze discuss this point and, based on the discussion, they could worthily extent the scope/applicability of their proposed study.

Minor

·         The authors should check that all the used acronyms are explained.

·         The authors could evaluate to move some tables in Appendices, for the sake of readability.

Author Response

Please find the responses in the attached file.

Reviewer 2 Report

The paper presents climate data to undertake hygro-thermal and building simulations under projected climate change influences for some chosen cities in North America. Authors predict that buildings in Canada will be exposed to strong climatic conditions in coming years as result of climate change. They propose to evaluate buildings performance. Authors say that building simulation models need continuous climate records of following parameters: solar radiation, cloud-cover, wind, humidity, rainfall, temperature, and snow-cover.

Chapter 1 – Summary? – should have different name – Introduction. Please think about it. The chapter is well organized and introduce a reader to the topic of the paper. Authors present the history data regarding the climatic changes for last decades in Canadian cities. They also explain how the data were collected.

Chapter 2 – methods – shows selected 11 cities. The chapter describes climatic variables, such as horizontal and normal irradiance, cloud cover, rainfall, wind direction, temperature, others. Next subchapter presents data sources, such as climate observations, simulations. The chapter describes also variables, like rainfall, snow cover.

Chapter 3 – presents the analysis. It shows validation of the procedure, means the comparison between model results and real data.

I think, the paper results are able to help to include climate changes during building procedure in Canada. Presented procedure can be also valid and use in other countries, which do not have such a project yet. I think the paper is ready to be published in present form.

Author Response

Please find responses in the attached file.
